# MAC: A Live Benchmark for Multimodal Large Language Models in Scientific Understanding

**Mohan Jiang**[1,2*], **Jin Gao**[1*], **Jiahao Zhan**[3], **Dequan Wang**[1,2†]
[1]Shanghai Jiao Tong University    [2]Shanghai Innovation Institute    [3]Fudan University

## Abstract

As multimodal large language models (MLLMs) become increasingly capable, fixed benchmarks are gradually losing their effectiveness in evaluating high-level scientific understanding. In this paper, we introduce the Multimodal Academic Cover benchmark (MAC), a live benchmark that could continuously evolve with scientific advancement and model progress. MAC leverages over 25,000 image-text pairs sourced from issues of top-tier scientific journals such as Nature, Science, and Cell, challenging MLLMs to reason across abstract visual and textual scientific content. Experiments on our most recent yearly snapshot, MAC-2025, reveal that while MLLMs demonstrate strong perceptual abilities, their cross-modal scientific reasoning remains limited. To bridge this gap, we propose DAD, a lightweight inference-time approach that enhances MLLM by extending visual features with language space reasoning, achieving performance improvements of up to 11%. Finally, we highlight the live nature of MAC through experiments on updating journal covers and models for curation, illustrating its potential to remain aligned with the frontier of human knowledge. We release our benchmark at https://github.com/mhjiang0408/MAC_Bench.

## 1 Introduction

With extensive research driving into developing multimodal large language models (MLLMs) for autonomous scientific research based on their exceptional capabilities in complex domains, evaluating MLLMs for scientific understanding remains a long-standing and evolving challenge. In earlier stages, benchmarks such as MMMU (Yue et al., 2024) served as demanding tests of multimodal reasoning, particularly in scientific domains. However, as MLLMs grow increasingly capable, such benchmarks show signs of saturation. For example, Gemini-2.5-pro (Google, 2025) recently achieved 81.7% accuracy in a pass@1 setting on MMMU, suggesting that even with comprehensive curation, static benchmarks may lose their guiding influence over time.

This phenomenon highlights the need for a *live* benchmark—one that evolves alongside model progress, rather than remaining static. Recent efforts like LiveBench (White et al., 2024) aim to address this by regularly releasing new questions sourced from dynamic content such as arXiv papers, news articles, and IMDb synopses. However, LiveBench remains unimodal, focusing solely on language, and its reliance on rapidly changing content from the Internet raises concerns about data quality in scientific understanding.

Leading scientific journals, carefully curated by experts, show great potential as they publish new issues weekly or monthly. Each issue contains a journal cover and a corresponding cover story, depicting the same scientific topic from vision and language modalities. These image-text pairs encapsulate complex scientific concepts through abstract multimodal representation. As cutting-edge discoveries, they often fall outside existing model pretraining corpora, making them ideal testbeds for probing scientific understanding in MLLMs.

Motivated by this, we introduce the Multimodal Academic Cover benchmark (MAC), a *live* benchmark built from *live* scientific journals and curated through our *live* curation mecha-

---

* Equal contribution. † Corresponding author: dequanwang@sjtu.edu.cn.

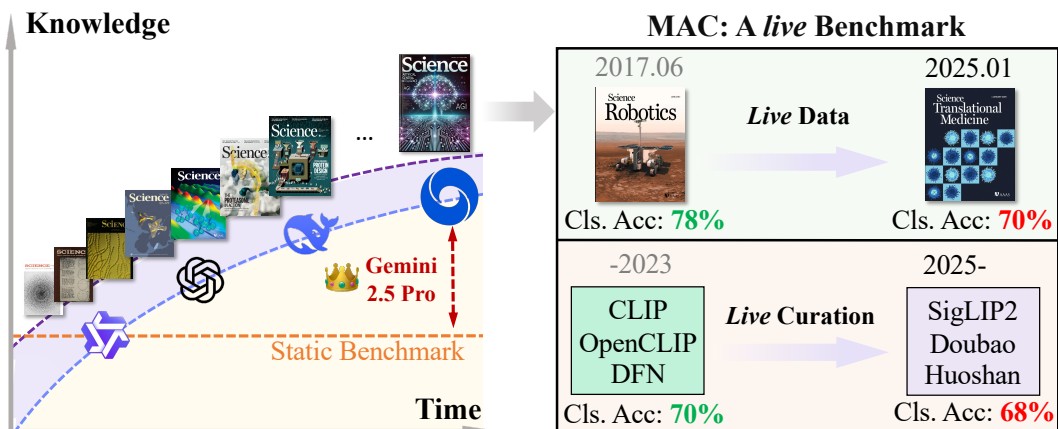

Figure 1: **MAC is a *live* benchmark to evaluate the scientific understanding of MLLMs.** By continuously incorporating the latest scientific discoveries and curating with the latest embedding models, our *live* benchmark transcends the limitations of static benchmarks that face progressive performance saturation by developing MLLMs. Best viewed in color.

nism to test MLLMs' scientific visual understanding and cross-modal concept alignment capabilities, shown in Figure 1. Each question presents a four-way classification task: given a cover image, select the corresponding cover story (Image2Text), or given a story, select the matching cover (Text2Image). Distractors are selected by embedding similarity across multiple models, ensuring the benchmark remains challenging even for the latest MLLMs.

We present MAC-2025, the most recent yearly snapshot of MAC, constructed from over 2,000 cover image-story pairs drawn from premier journals such as Nature, Science, and Cell, covering issues from January 2024 to February 2025. We evaluate seven leading MLLMs on both Image2Text and Text2Image tasks, revealing that while they exhibit strong visual recognition capabilities, such performance heavily relies on the text on cover images.

To better handle the challenging visual information in MAC-2025, we introduce Description and Deduction (DAD), a simple yet effective inference-time approach that enables MLLMs not only to perceive visual input, but also to understand and reason across modalities. DAD builds a two-stage cross-modal reasoning pipeline that first extracts multi-granularity visual features using an MLLM, and then applies a language model to perform high-level reasoning. This hybrid approach yields consistent gains of 1%–5% across baselines, with Ernie-4.5-8k (Baidu, 2025) achieving an 11% improvement, demonstrating its ability to equip MLLMs with greater intellectual capacity in scientific visual understanding.

Beyond the snapshot, we explore the *live* nature of MAC from two perspectives (Figure 1), *live* data and *live* data curation. To assess data evolution, we collect a large-scale dataset comprising over 25,000 image-text pairs from past and current journal issues. Our analysis shows that MLLMs perform better on older issues, underscoring the increasing complexity and novelty of recent scientific content. For benchmark construction, we regenerate MAC-2025 using the latest embedding models released in 2025 (concurrent to our work), which yield harder distractors and further degrade model accuracy, demonstrating our benchmark's adaptability to advances in representation learning.

Our contributions are shown as follows:

- We introduce MAC, a continuously updating benchmark for evaluating multimodal scientific understanding in MLLMs. Based on live scientific journals, our benchmark is built using a live mechanism that adapts to model progress.

- We study the latest yearly snapshot of MAC, MAC-2025, drawn from over 2,000 curated journal issues, and provide a thorough evaluation of seven advanced MLLMs on both Image2Text and Text2Image tasks.

- We propose an inference-time approach, Description and Deduction (DAD). It significantly enhances MLLMs' scientific concept reasoning by bridging cross-modal information between MLLMs and a reasoning language model.

- We investigate the live attribute of our MAC through temporal data analysis and adaptive benchmark construction, showing the necessity of growing scientific journals and evolving construction using embedding models.

## 2 Related Work

**Multimodal Large Language Model**   Since the foundational work of Radford et al. (2021) on joint image-text representations, multimodal large language models (MLLMs) have rapidly advanced, leading to a series of notable models such as Zeng et al. (2022), Driess et al. (2023), Achiam et al. (2023), Liu et al. (2023), and GLM et al. (2024). Recent progress has shown their strong potential across a wide range of tasks and applications (Huo et al., 2024; Xi et al., 2025). Proprietary models like GPT-4o (Hurst et al., 2024) and Claude3.5 (Anthropic, 2024) have achieved outstanding results on benchmarks (Song et al., 2024; Wang et al., 2024b), while open-source models such as LLaVA-NeXT (Liu et al., 2024a), DeepSeek-VL (Lu et al., 2024), InternVL 2.5 (Chen et al., 2024b) and Qwen2.5-VL (Bai et al., 2025) leverage advanced projection techniques to integrate visual and textual features for efficient multimodal understanding. These developments underscore the growing impact of MLLMs in both research and real-world applications. In this context, we introduce Multimodal Academic Cover benchmark (MAC)—a benchmark specifically designed to assess MLLMs' ability to comprehend implicit scientific concepts in cover images.

**MLLM benchmark**   With the rapid advancement of MLLMs, many benchmarks have emerged, highlighting the importance of evaluating both perception-understanding and cognition-reasoning capabilities. Liu et al. (2024b) focused on basic perception tasks like object localization via multiple-choice questions, while Wang et al. (2024a), Meng et al. (2024), and Kil et al. (2024) assessed multi-image understanding and comparative reasoning through complex visual tasks. Lu et al. (2023) and Cao et al. (2024) examined logical reasoning using abstract visual questions and mathematical problems. Cross-domain knowledge was evaluated by Lu et al. (2022), Zhang et al. (2023), and Yue et al. (2024) using multidisciplinary questions across educational levels. However, most existing benchmarks are static and struggle to remain challenging as MLLMs rapidly evolve (Bai et al., 2025; Lu et al., 2024; Chen et al., 2024b), lacking mechanisms for regular updates or scalable expansion. In contrast, our MAC offers a sustainable evaluation framework that maintains appropriate difficulty while accurately measuring scientific comprehension, enabling continuous assessment of MLLMs' scientific visual understanding and cross-modal concept alignment capabilities.

**Scientific Figure Question-Answering Benchmark**   Evaluating MLLMs' ability to understand scientific figures is critical for assessing their deeper reasoning skills, where Li et al. (2025) released concurrent to our work reinforces this importance by incorporating scientific knowledge into generative models through Science-T2I and SciScore. Early benchmarks (Kahou et al., 2017; Chen et al., 2020) focused on synthetic chart-based VQA datasets (*e.g.*, line and bar graphs), emphasizing visual parsing over scientific understanding. Later efforts (Masry et al., 2022; Methani et al., 2020; Li & Tajbakhsh, 2023; Hu et al., 2024; Li et al., 2024a; Pramanick et al., 2024; Li et al., 2024b) introduced more realistic figures, from real-world scenarios or arXiv papers across disciplines. However, these datasets typically rely on figure-caption pairs, where captions may omit key scientific insights encoded in the visual data, limiting the evaluation to surface-level format comprehension. Moreover, the quality and completeness of captions can vary significantly, introducing noise and inconsistency. MAC-2025 addresses this gap by using covers and cover stories from 188 journals across four major publishers. Crafted by professional artists and editors, these pairs exhibit strong semantic alignment, embedding rich scientific meaning in both text and imagery. This enables a more rigorous test of MLLMs' ability to interpret high-level scientific visual concepts.

## 3 Benchmark

### 3.1 Dataset Collection and Structure

To support a comprehensive benchmark, we first present a large-scale repository of 188 academic journals, including main and subsidiary titles from leading scientific journal presses from their inception to February 2025. Each issue is represented as a tuple of (1) *Cover Image*, which displays the front cover of the issue, and (2) *Cover Story* that explains the scientific concept behind the cover and introduces the featured article. To keep the benchmark aligned with the cutting edge of scientific knowledge and challenging for current MLLMs, we construct MAC-2025, the latest yearly snapshot of MAC from January 2024 to February 2025, which serves as the primary focus of our empirical analysis.

The journal collection in MAC is sourced from the official websites of four renowned journal series, which are briefly introduced below.

- **Cell (Cell, 2025) and its sub-journals**: MAC includes 42 sub-journals from the Cell series, which primarily focus on molecular and cell biology, comprising a total of 7,619 issues;
- **Nature (Nature, 2025) and sub-journals**: This segment includes 67 journals from the Nature series, known for its multidisciplinary coverage across fields such as materials, pharmaceuticals, energy, and genetics.
- **Science (Science, 2025) and its sub-journals**: MAC includes all six journals in the Science family, totaling 2,792 issues, covering interdisciplinary research across physics, medicine, and engineering.;
- **American Chemical Society (ACS) (ACS (2025)) Publications**: This segment includes 73 journals with 8,004 issues, covering core chemistry fields and interdisciplinary areas such as materials science and chemical engineering.

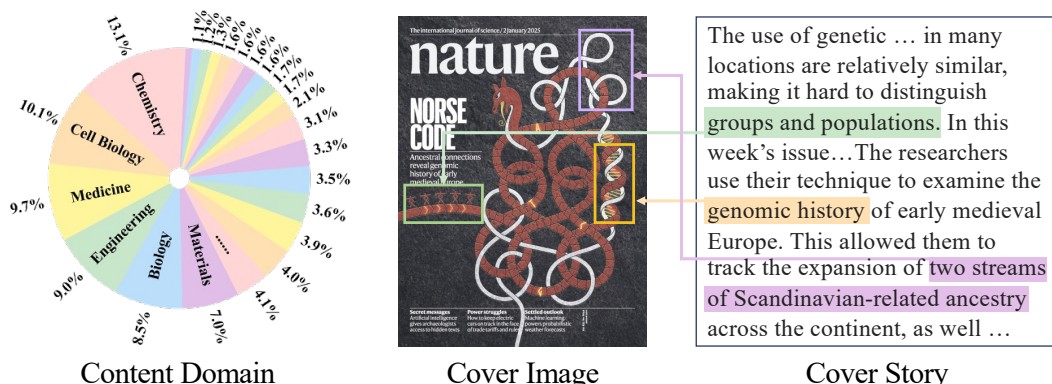

Content Domain          Cover Image          Cover Story

Figure 2: **Our MAC spans diverse scientific disciplines, offering comprehensive visual and textual scientific materials.** *Left* presents the distribution of disciplinary categories, while *right* demonstrates representative examples of visual-textual scientific concept alignments from our dataset. Best viewed in color.

MAC comprises cover-story pairs from authoritative scientific journals known for their high editorial standards and visually compelling, content-aligned covers. Each issue includes a professionally written cover story conveying rich scientific concepts (Figure 2), often expressed in complex, domain-specific language, posing challenges for MLLMs. We collect these pairs from open-access sources and remove duplicates with identical images.

To address the rapid saturation of existing benchmarks, we curate a recent snapshot—MAC-2025 —containing 2,287 samples from discoveries published between January 2024 and February 2025. These cutting-edge samples, rarely seen in training data, allow for a more accurate and forward-looking evaluation of MLLMs' scientific understanding. To overcome

the limitations of static benchmarks, we adopt an annual release schedule with each year's latest snapshot released every March, aligned with journal publication cycles, enabling dynamic updates that reflect both scientific and model evolution. MAC-2025 thus serves as a sustainable, evolving benchmark that maintains challenging standards while keeping pace with MLLM progress.

## 3.2 MAC-2025

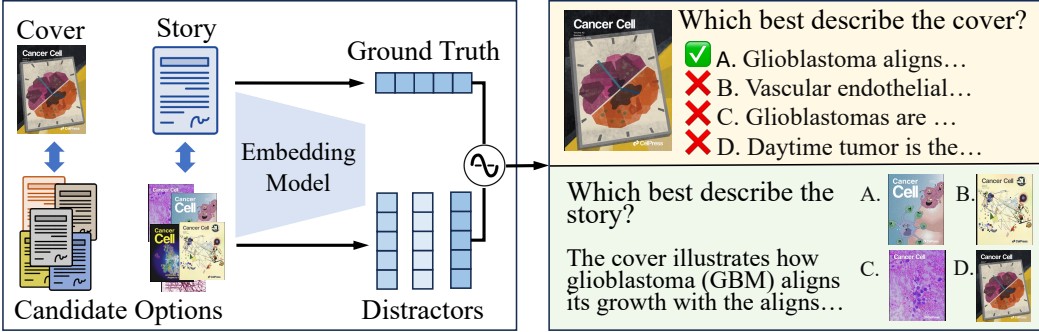

Figure 3: **Construction pipeline and benchmark examples.** Our benchmark incorporates four distinct evaluation tracks, leveraging two categories of semantic similarity comparisons through embedding model representations and bidirectional understanding tasks encompassing both image-to-text and text-to-image understanding. Best viewed in color.

We design two modality-specific tasks shown in Figure 3 based on the cover images and stories to assess the performance of MLLMs in processing multimodal information and achieving target objectives. Our tasks are presented in the form of classification questions.

**Bidirectional Tasks** We evaluate the model through two bidirectional multimodal tasks, assessing its ability to perceive visual elements in images and to understand the scientific concepts embedded in both text and visuals, and thereby examine its performance in cross-modal semantic alignment and reasoning. The Image2Text task is selecting cover stories given cover images. We first present MLLMs with a journal cover and ask them to select the most relevant cover story from four options. The Text2Image task is selecting cover images given cover stories. MLLMs are required to identify the cover image, among four candidates, that corresponds most accurately to the provided cover story.

**Curation of Distractors** For Image2Text and Text2Image classification tasks, distractors are drawn from cover stories within the same journal to ensure similar knowledge domains. We introduce two selection methods to maximize confusion for the models from two domains, shown in the left part of Figure 3. The information domain (info domain) means selecting distractors based on a higher similarity to the information provided in the task. The option domain (option domain) means selecting distractors based on a higher similarity to the option that serves as the ground truth in the task.

To build similarity metrics for the info and option domains (Li et al., 2025), we select three embedding models per modality and compute cosine similarities. For text-only comparisons (Image2Text option domain), we use three SentenceTransformer models (Reimers & Gurevych, 2020), multi-qa-mpnet-base-dot-v1, all-mpnet-base-v2, and paraphrase-MiniLM-L6-v2; for image or cross-modal comparisons (Text2Image full task and Image2Text info domain), we use three multimodal visual encoders, clip-vit-large-patch14-336 (Radford et al., 2021), ViT-g-14 (Ilharco et al., 2021), and ViT-H-14-quickgelu (Fang et al., 2023). These models, trained on diverse datasets with varying scales, provide complementary strengths. For each classification question, we select the top-ranked distractor from each model and choose the final one based on the highest average rank when overlaps occur.

| MLLMs | Image2Text Level | | | | Text2Image Level | | | |
| | Info Domain | | Option Domain | | Info Domain | | Option Domain | |
| | Acc.(%)↑ | ECE ↓ | Acc.(%)↑ | ECE ↓ | Acc.(%)↑ | ECE ↓ | Acc.(%)↑ | ECE ↓ |
|---|---|---|---|---|---|---|---|---|
| Qwen2.5-VL-7B | 59.7 | 0.055 | 60.5 | 0.063 | 57.1 | 0.120 | 61.0 | 0.131 |
| Qwen-VL-Max | 69.8 | 0.171 | 69.6 | 0.180 | 70.4 | 0.214 | 72.6 | 0.232 |
| Step-1V-8k | 67.6 | 0.181 | 69.0 | 0.138 | 64.4 | 0.124 | 65.1 | 0.117 |
| Step-1o-T-V | 70.4 | 0.061 | 68.7 | 0.079 | 69.6 | 0.083 | 71.5 | 0.116 |
| Ernie-4.5-8k | 57.5 | 0.231 | 61.4 | 0.158 | 55.8 | 0.095 | 58.3 | 0.053 |
| Gemini-1.5-Pro | 72.7 | 0.108 | 70.4 | 0.085 | 71.8 | 0.237 | 72.8 | 0.172 |
| GPT-4o | 75.1 | **0.053** | 73.5 | **0.055** | 74.3 | **0.038** | 74.3 | 0.068 |
| Step-3 | **77.3** | 0.069 | **75.4** | 0.057 | **78.0** | 0.065 | **79.1** | **0.048** |

Table 1: **Performance on MAC-2025.** The table presents the experimental results of MLLMs on the understanding tasks of MAC-2025, with **bold** indicating the best results and underlined indicating the second-best results. Step-1o-T-V represents Step-1o-Turbo-Vision.

**Self-Validation Experiment**   While we use embedding similarity to select challenging distractors, this raises two concerns: whether embedding models can solve our bidirectional matching tasks using similarity alone, and whether the scores truly reflect semantic alignment. To explore this, we conduct self-verification experiments across both modalities (Table 2). Results show that although embedding models achieve under 40% accuracy—revealing their limitations in cross-modal understanding—their self-rankings fall within the top 25%, suggesting the selected distractors are semantically close and effectively challenging. Additional analysis is discussed in section B and section F.

### 3.3   Description and Deduction

We first experiment with traditional in-context learning methods and find that they are ineffective for complex scientific understanding tasks (Table 4 and Table 5). Motivated by the inference-time scaling Snell et al. (2024) and a series of powerful inference models, such as ChatGPT-o1 (OpenAI, 2024), QwQ-32B (Team, 2025), and DeepSeek-R1 (Guo et al., 2025), we propose **Description and Deduction (DAD)**, a two-stage inference-time approach specifically designed to enhance MLLMs' comprehension of scientific cover images.

In the first stage, the full cover image and its classification question are fed into an MLLM to generate a detailed image description and a pseudo chain-of-thought (Huang et al., 2025), which guides reasoning without revealing the answer. Crucially, both DAD and MLLM approaches process all images simultaneously in a single batch rather than sequentially, ensuring that any performance variations stem solely from the introduction of the reasoning model, thereby maintaining evaluation fairness. In the second stage, these outputs are combined with the question text and passed to a dedicated reasoning model that produces a probability distribution over the options. By bridging visual perception with high-level reasoning, this structured process improves interpretability and equips MLLMs with the ability to reason about deeper connections between visual elements and scientific concepts.

## 4   Experiments

We first introduce the baseline models and evaluation metrics (section 4.1), followed by the understanding evaluation of these state-of-the-art models and the performance gains achieved through our Description and Deduction framework on MAC-2025 (section 4.2). We next assess the effectiveness of both baseline models and our method on the text-free version of MAC-2025 (section 4.3). Finally, we analyze the impact of different reasoning models integrated into our framework (section 4.4).

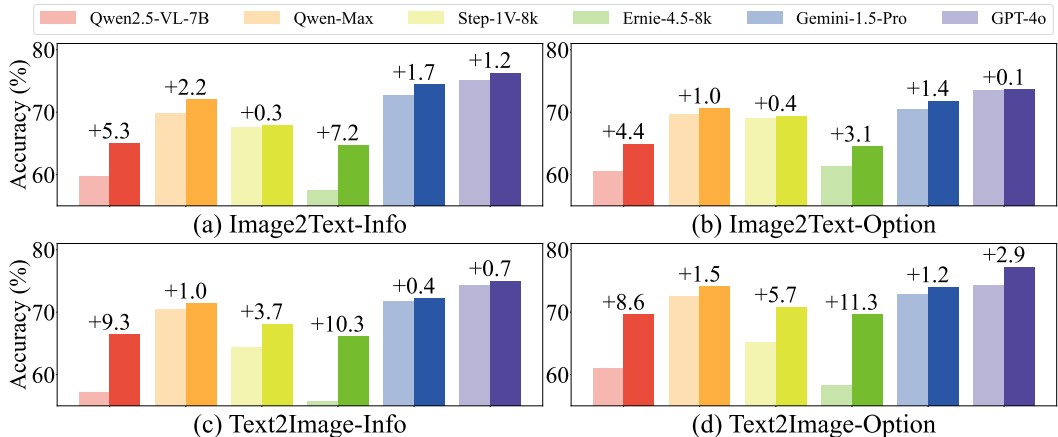

Figure 4: **Improvements of our DAD across MLLMs.** Different colors represent different models, with the darker bars next to each model indicating the effect of our method. DAD is generally effective in all four settings, shown in Subfigures (a)-(d). Best viewed in color.

## 4.1 Settings

**Models**  We start by reviewing both open-source and closed-source multimodal large language models (MLLMs), finding that closed-source models consistently outperform their open-source counterparts across multiple benchmarks (Guan et al., 2024; Yue et al., 2024; Chen et al., 2024a; Su et al., 2025). In light of the observed performance gap, our baselines primarily include leading commercial closed-source models: Qwen-VL-Max (Team, 2024b), Step-1V-8k (StepFun, 2024b), GPT-4o (Hurst et al., 2024), Gemini-1.5-Pro (Team et al., 2024), and Ernie-4.5-8k (Baidu, 2025). For comparison, we also report results of the open-source model Qwen2.5-vl-7B (Bai et al., 2025) and the visual reasoning model Step-1o-Turbo-Vision (StepFun, 2024a). To reflect the latest advances in multimodal models, we also evaluate Step-3, the strongest open-source multimodal reasoning model concurrent to our work. We leverage QwQ-32B (Team, 2025) as the reasoning model in Description and Deduction (DAD) to evaluate the effects of inference-time scaling.

**Metrics**  In our experiments, we report 4 metrics of each MLLM on MAC-2025 to reflect the models' classification performance and confidence performance on our benchmark:

- **Accuracy (Acc.)** measures the proportion of correct predictions, reflecting models' ability to identify correct scientific concepts.
- **Expected Calibration Error (ECE)** (Guo et al., 2017) evaluates prediction reliability, measuring whether models' confidence levels match their actual accuracy.
- **Negative Log-Likelihood (NLL)** (Guo et al., 2017) assesses prediction quality, penalizing both incorrect predictions and misaligned confidence, with higher penalties for high-confidence errors.
- **Root Mean Square Error (RMS)** (Hendrycks et al., 2018) quantifies prediction error, measuring the magnitude of prediction deviations.

## 4.2 Understanding Evaluation

Table 1 and Figure 4 present classification results under four experimental settings, where models must identify visual scientific elements and extract relevant scientific concepts, aligning them with concepts in corresponding cover stories. We observe that these state-of-the-art MLLMs achieve accuracy rates between 50% and 80% on our MAC-2025, highlighting the benchmark's strong discriminative capacity in evaluating models' comprehension of complex, cutting-edge scientific content. This suggests that existing models still face notable challenges in processing multimodal information and performing higher-level scientific

reasoning. However, our proposed method DAD, whose result is shown in Figure 4, serving as a slight-weight inference-time approach, significantly improves the performance of these MLLMs, particularly achieving an 11.3% accuracy gain for Ernie-4.5-8k in the option domain of the Text2Image task. Through effective cross-modal information fusion and optimization of reasoning mechanisms, our approach enhances each model's understanding and reasoning capabilities regarding scientific concepts, demonstrating robust performance.

## 4.3 Performance on Text-Free Cover Image

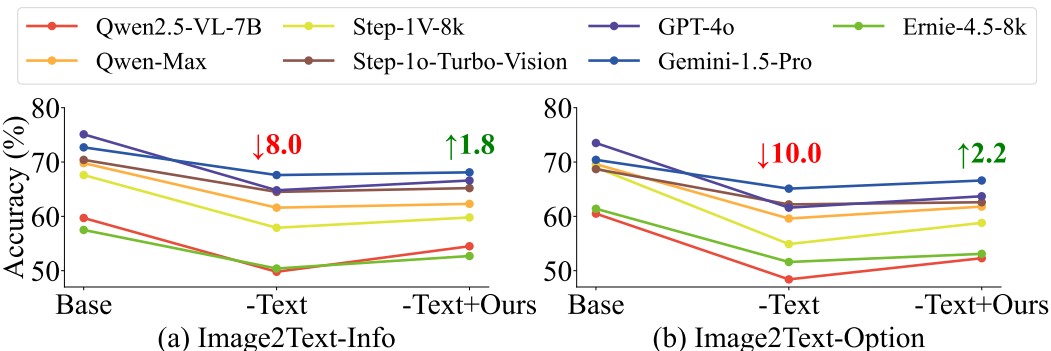

(a) Image2Text-Info        (b) Image2Text-Option

Figure 5: **Performance on covers with or without texts.** "-Text" indicates the removal of text from cover images using OCR on MAC-2025. "+Ours" refers to our DAD instead of simple MLLMs. Removing text leads to a significant performance drop, while our DAD approach remains robust, even without relying on textual content. Best viewed in color.

Since MLLMs may rely on rich textual elements, such as journal titles, author names, and detailed annotations, rather than visual content to comprehend scientific cover images, we design a controlled experiment to evaluate their ability to understand scientific concepts based solely on visual information. To this end, we systematically apply OCR (JaidedAI, 2021) to the original images to identify and remove the main textual regions, subsequently masking them with white blocks to eliminate potential textual cues while preserving the overall visual structure. We set the confidence threshold for this process to 0.25. This value is determined by experimenting with thresholds ranging from 0.1 to 0.9 on 10 randomly sampled cover images from four different publishers. The final value is chosen by considering both the effectiveness of text removal and the preservation of key visual-semantic elements in the images. We then assess several representative models under this text-free setting and conduct an ablation experiment where text is removed from images but provided in the prompt to determine whether performance degradation is due to image distortion.

As shown in Figure 5, removing text causes a notable drop in accuracy, revealing that current models struggle to reason about complex scientific concepts from visual input alone, while Table 6 indicates that the performance degradation is not due to image distortion. The case study in Figure 8 demonstrates that even the most advanced model released concurrent to our work, Gemini 2.5 Pro (Google, 2025), struggles to reason from perceived visual elements to complex scientific concepts. In contrast, our DAD method significantly improves performance by leveraging test-time scaling and relying solely on the initial visual grounding step. These results indicate that while MLLMs excel at perceiving elements in cover images, their visual-only scientific understanding remains limited, yet can be effectively enhanced through structured reasoning mechanisms.

## 4.4 Reasoning Module Comparison in DAD

To further assess the impact of different reasoning models within our Description and Deduction framework, we replaced the original QwQ-32B reasoning model with several representative alternatives, including Deepseek-R1 (Guo et al., 2025) and ChatGPT-o3-mini-high (OpenAI, 2025). As shown in Table 7, the results reveal notable performance differences

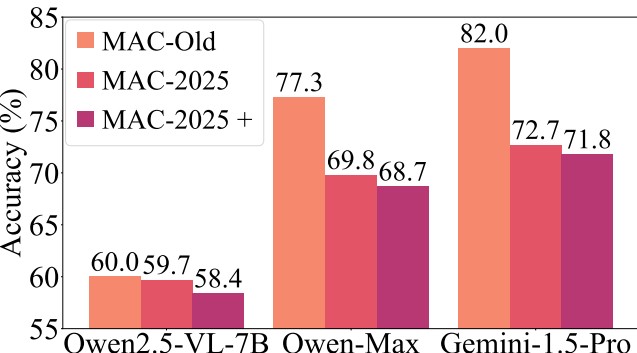

Figure 6: **Benchmark with distinct data and models for curation across different periods.** MAC-Old and MAC-2025 use the oldest and latest journal issues, respectively. "MAC-2025 +" is curated by the latest embedding models. Our MAC is getting more challenging, benefited from *live* data and *live* data curation.

across reasoning models. Incorporating more advanced models, such as Deepseek-R1, leads to significant accuracy gains, even for already strong MLLMs. The integration of reasoning models via the DAD method consistently yields high accuracy and robustness in image-to-text tasks. These findings underscore the importance of inference-time scaling in multimodal understanding and offer valuable insights for optimizing future MLLMs.

## 5 MAC's *Live* Attribute

To rigorously demonstrate the *live* attribute of our Multimodal Academic Cover benchmark, we conduct experiments on two perspectives, *live* data and *live* data curation. In section 5.1, we demonstrate that newer data, represented by the MAC-2025 subset, poses significantly greater challenges than earlier subsets, underscoring the increased complexity introduced by cutting-edge scientific knowledge. In section 5.2, we further amplify the benchmark's difficulty by curating distractors using contemporary embedding models. Our controlled experiments show that stronger embeddings yield more confusing distractors, resulting in substantial accuracy drops even for state-of-the-art MLLMs.

### 5.1 *Live* Data

To demonstrate that recent data poses greater challenges to current MLLMs, we evaluate model performance on over 2,000 samples drawn from the earliest available year across all journals in MAC, shown as the MAC-Old in Figure 6. Comparative analysis shows that models perform significantly worse on MAC-2025 than on earlier subsets MAC-Old, indicating that incorporating newer, cutting-edge scientific discoveries substantially increases task difficulty (Table 3). This temporal performance degradation can be attributed to the fact that recent scientific advances often introduce novel concepts, methodologies, and terminology that fall outside the training distributions of current models, creating inherent challenges for comprehension and reasoning. This notable temporal performance gap further supports our *live* design philosophy: by continuously integrating the latest scientific advances, the benchmark preserves both its relevance and difficulty. Our *live* MAC-2025 leverages up-to-date frontier scientific knowledge to evaluate models' scientific understanding, ensuring consistent and fine-grained evaluation of their capabilities.

### 5.2 *Live* Data Curation

To reveal how evolving embedding models enhance benchmark's challenging nature, we rebuilt distractors of the classification questions in MAC-2025 using three contemporary embedding models introduced concurrently with our work—SigLip2 (Tschannen et al., 2025), Qwen-Multimodal-Embedding-V1 (Team, 2024a), and doubao-embedding-vision-241215 (Doubao, 2024)—while keeping the original information set fixed. This controlled experimental setup allows us to isolate the impact of updated distractors on evaluation difficulty while maintaining consistency in the underlying knowledge being tested. The selection of these three embedding models represents newer and more powerful itera-

tions that demonstrate significant improvements over previous generations, ensuring that our analysis captures how enhanced embedding capabilities influence distractor quality. Furthermore, by maintaining identical correct answers and question structures across all experimental conditions, we establish a rigorous framework for attributing performance variations solely to the sophistication of the distractor generation process. Experiments on the Image2Text task with three baseline models (see results in Figure 6 and details in Table 8) show that stronger embedding models produce more confusing distractors, leading to notable accuracy drops and less reliable predictions.

These findings have several important implications: First, they validate our hypothesis that more advanced embedding models can generate more challenging distractor items; Second, the performance degradation indicates that even state-of-the-art MLLMs still have limitations in processing subtle semantic differences; Finally, this approach provides new insights into constructing dynamically adaptive benchmarks that evolve alongside model capabilities. Our *live* MAC-2025 utilizes evolving multimodal embedding models to construct distractors matched to the current capabilities of MLLMs, ensuring sustained challenge and reliability while offering valuable insights for future benchmark design and optimization.

## 6 Conclusion

In this work, we introduce Multimodal Academic Cover benchmark, a live benchmark that dynamically evolves alongside scientific advances and model progress to evaluate MLLMs' scientific understanding capabilities. Leveraging over 25,000 carefully curated image-text pairs from leading scientific journals and concentrating on our latest snapshot MAC-2025, we reveal a critical finding: while current MLLMs demonstrate strong visual perception abilities, they exhibit significant limitations in cross-modal scientific reasoning. To address this fundamental gap, we propose Description and Deduction, a lightweight inference-time approach that effectively bridges visual cues with language-based reasoning processes, achieving substantial performance improvements across multiple scientific domains. Through rigorous temporal generalization analysis and systematic evolving benchmark construction, we further demonstrate the necessity and feasibility of live evaluation frameworks, highlighting the critical importance of continuously adaptive benchmarks for accurately monitoring MLLMs' scientific comprehension capabilities as they advance over time.

Looking forward, we envision several promising directions for advancing scientific MLLM evaluation. A key priority lies in developing more fine-grained evaluation metrics that can systematically distinguish between perception-understanding and cognition-reasoning abilities. Specifically, we plan to design granular assessment frameworks that isolate visual perception tasks such as identifying scientific items and extracting textual information from higher-order cognitive reasoning tasks including causal inference, hypothesis formation, and scientific concept understanding. This differentiation will enable more precise diagnosis of model capabilities and targeted improvements in specific areas. Additionally, we aim to expand our evaluation paradigm beyond discriminative tasks to include generation-based assessments, where models must produce coherent scientific explanations and demonstrate reasoning processes. We believe that such comprehensive and evolving benchmarks will not only provide deeper insights into current model limitations but also serve as crucial catalysts for the development of more sophisticated MLLMs, ultimately driving the field toward systems capable of genuine scientific discovery and reasoning.

## Acknowledgement

This research is supported by the Key R&D Program of Shandong Province, China (2023CXGC010214, 2024CXGC010213). We express our gratitude to the funding agency for their support. We thank all the anonymous reviewers for their valuable suggestions.

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

## A  Open Access License of the Journals

All cover images and their corresponding cover stories used in the construction of Multi-modal Academic Cover benchmark (MAC) are sourced from the official websites of leading scientific journals which published by Nature Portfolio, Science/AAAS, Cell Press and American Chemical Society. We strictly ensure that all content included in the MAC complies with the open access or public use policies stated by each publisher.

The use of cover images and their associated cover stories is strictly limited to non-commercial academic research purposes. We do not redistribute full journal articles, nor do we modify, alter, or claim ownership of any original content. Proper attribution and source referencing are preserved wherever applicable, and the entire dataset construction process adheres to established fair use principles, with the goal of supporting rigorous scientific evaluation, reproducibility, and transparency in academic research.

## B  Self-Validation Experiment Result

| Modal | Embedding Model | Average↓ | Median↑ | Acc.(%)↑ |
|---|---|---|---|---|
| Image2Text | CLIP-L | 0.255 | 0.150 | 16.2 |
| | ViT-H | 0.227 | 0.105 | **23.9** |
| | ViT-G | 0.287 | 0.193 | 16.7 |
| | average | 0.237 | 0.128 | 21.6 |
| Text2Image | CLIP-L | 0.203 | 0.044 | 32.0 |
| | ViT-H | 0.205 | 0.037 | 35.8 |
| | ViT-G | 0.238 | 0.076 | 29.2 |
| | average | 0.194 | 0.034 | **38.7** |

Table 2: **Analysis of multimodal Embedding Models' Performance in Scientific Understanding Tasks and Distractor Selection.** We assessed the models through two metrics: self-ranking (similarity rankings of paired stories or images among all journal samples) and accuracy rates in scientific comprehension multiple-choice tasks.

Table 2 presents the self-validation results of the embedding models used in selecting distractors for MAC-2025. Specifically, "Average" indicates the average relative rank of self-similarity for each embedding model across all journal issues—that is, the rank of similarity between the embedding of an image and its corresponding story, relative to all other candidate stories from the same journal. Median represents the median of these self-similarity rankings. Acc. refers to the model's accuracy in answering the finalized MAC-2025 classification questions using similarity alone. The "average" row under each modality corresponds to a combined ranking, obtained by averaging the ranks produced by the three embedding models for each issue and re-ranking accordingly, thus reflecting a composite similarity measure that integrates their respective strengths.

We evaluate the capability of three embedding models (CLIP-L, ViT-H, and ViT-G) on our bidirectional matching tasks using similarity-based retrieval, as reported in Table 2. Results reveal a consistent performance gap between the Image2Text and Text2Image tasks. Specifically, the average accuracy across models for Image2Text is 21.6%, with an average rank of 0.237, whereas Text2Image achieves a substantially higher average accuracy of 38.7% and a lower average rank of 0.194. This indicates that retrieving matching images from text is generally easier than retrieving text from images, likely due to the higher semantic complexity embedded in scientific visual content. Among the models, ViT-H consistently outperforms others across both modalities, achieving the highest accuracy in Image2Text (23.9%) and Text2Image (35.8%), along with competitive average ranks (0.227 and 0.205, respectively). In contrast, ViT-G performs the worst, with lower accuracy and higher rank values in both tasks, suggesting limited effectiveness in capturing cross-modal semantic alignment. Despite the relatively low accuracy scores—particularly in the Image2Text

setting—all models rank the correct answer within the top 25% on average, indicating that the selected distractors are semantically close and thus effective for challenging MLLMs.

## C  Human Evaluation

To evaluate human performance on our benchmark, we selected 20 journal covers from diverse publishers in the MAC-2025 dataset and designed two evaluation configurations: (1) random distractors and (2) embedding-based distractors consistent with the image2text info domain methodology employed in MAC-2025.

We recruited five independent evaluators, each with at least a bachelor's degree, to complete the assessment tasks. The results demonstrate notable performance differences: human evaluators achieved 80% accuracy with random distractors, while accuracy decreased to 76% with embedding-based distractors. This performance degradation provides empirical evidence for the effectiveness of our distractor design strategy. The significant accuracy reduction when using semantically-informed distractors validates our approach and confirms that the embedding-based methodology successfully increases task difficulty.

## D  MAC-Old Benchmark Evaluation

| MLLMs | Image2Text Level | | | | Text2Image Level | | | |
|---|---|---|---|---|---|---|---|---|
| | info domain | | option domain | | info domain | | option domain | |
| | Acc.(%)↑ | ECE↓ | Acc.(%)↑ | ECE↓ | Acc.(%)↑ | ECE↓ | Acc.(%)↑ | ECE↓ |
| Qwen2.5-VL-7B | 60.0 | **0.042** | 62.1 | 0.068 | 59.4 | 0.126 | 61.7 | 0.147 |
| Qwen-VL-Max | 77.3 | 0.196 | 75.0 | 0.196 | 74.5 | 0.244 | 75.5 | 0.273 |
| Gemini-1.5-Pro | **82.0** | 0.149 | **80.6** | 0.144 | 72.0 | 0.141 | 76.0 | 0.202 |
| GPT-4o | 77.6 | 0.051 | 74.8 | **0.058** | **76.2** | **0.071** | **79.3** | **0.083** |

Table 3: **MLLMs' Performance on Benchmarks Generated Using MAC-Old Dataset.** On MAC, four representative MLLMs achieve significantly higher accuracy compared to their performance on MAC-2025, along with notably better confidence calibration.

All evaluated models achieve consistently higher accuracy on the MAC-Old, which consists of journal issues furthest from the present, compared to their performance on the more recent MAC-2025 dataset. Moreover, the Expected Calibration Error (ECE) is generally lower and more stable. For instance, in the info domain of image2text, GPT-4o achieves an accuracy of 0.751 and an impressively low ECE of 0.053 on the MAC-Old data. However, on the MAC-2025 dataset, its accuracy drops to 0.643 while the ECE rises to 0.094, reflecting a notable decline in both performance and calibration. This trend is broadly consistent across other models. Gemini-1.5-Pro achieves 71.8% accuracy in the text2image info task on MAC-Old, but only 65.9% on Latest; its ECE also increases from 0.212 to 0.237, indicating higher confusion and difficulty in interpreting more recent scientific content.

The scientific discoveries presented in the latest dataset are more cutting-edge and linguistically complex, thereby posing greater and more nuanced challenges for MLLMs in both scientific semantic comprehension and cross-modal reasoning. These findings support our proposed design principle of a *live* benchmark: by continually incorporating the latest scientific advances, the benchmark remains aligned with the frontier of human knowledge while sustaining its evaluative strength against powerful models.

| MLLMs | Image2Text Level | | | |
|---|---|---|---|---|
| | Acc.(%)↑ | ECE↓ | NLL↓ | RMS↓ |
| Qwen2.5-VL-7B | 59.7 | 0.055 | **1.856** | **0.095** |
| + CoT | 56.6 | 0.184 | 2.071 | 0.223 |
| + One-Shot | 43.2 | **0.123** | 4.136 | 0.155 |
| + Self-Consistency(5) | 59.2 | 0.214 | 2.897 | 0.285 |
| + ours | **65.0** | 0.127 | 2.729 | 0.162 |
| Qwen-VL-Max | 69.8 | 0.171 | 3.054 | 0.210 |
| + CoT | 69.8 | **0.070** | 1.517 | **0.115** |
| + One-Shot | 63.1 | 0.231 | 5.378 | 0.315 |
| + Self-Consistency(5) | 69.8 | 0.319 | 2.143 | 0.373 |
| + ours | **72.0** | 0.082 | 2.719 | 0.117 |

Table 4: **MLLMs' Performance on Benchmarks Generated Using State-of-the-Art Embedding Models.** We reconstructed new distractor items for samples identical to those in MAC-2025 from the MAC dataset using three state-of-the-art embedding models, and evaluated baseline models' performance on the image2text task.

# E   In-Context Learning

We further experimented with In-Context Learning strategies on two representative models, Qwen2.5-VL-7B and Qwen-Max, incorporating Chain-of-Thought (CoT), one-shot prompting, and self-consistency to enhance MLLM performance. Following the setups from Wei et al. (2022) and Wang et al. (2022), we implemented CoT and 5-sample self-consistency baselines, with results summarized in Table 4 for the Image2Text task.

The results show that CoT offers some benefit for Qwen-Max, significantly reducing NLL to 1.517 and RMS to 0.115, indicating improved confidence calibration. However, for Qwen2.5-VL-7B, CoT not only fails to improve accuracy but also increases ECE (from 0.055 to 0.184) and NLL (from 1.856 to 2.071), suggesting that CoT may introduce instability in weaker models. One-shot prompting yielded unstable results for this task. While it slightly improved ECE in some cases, it consistently led to substantial drops in accuracy for both models and dramatic increases in NLL and RMS. For example, Qwen-Max's NLL rose sharply to 5.378, indicating the limited generalization capacity of one-shot learning in complex scientific contexts. Self-consistency showed marginal improvement over few-shot on Qwen2.5-VL-7B, with accuracy reaching 0.592, though at the cost of higher ECE and RMS compared to the original model. On Qwen-Max, self-consistency failed to improve accuracy and even degraded calibration, with ECE increasing to 0.319.

In contrast, our proposed Description and Deduction consistently delivered notable and robust gains. Qwen2.5-VL-7B's accuracy improved from 0.597 to 0.650, and Qwen-Max from 0.698 to 0.720, while maintaining low ECE and RMS throughout. These results affirm the robustness and effectiveness of our structured reasoning mechanism in tracking complex scientific visual tasks and underscore the importance of incorporating explicit reasoning processes to endow MLLMs with deeper scientific understanding.

# F   Distractor Preference

As described in section 3.2, for the selection of distractors in the multiple-choice questions of MAC-2025, we introduced various embedding models trained on different datasets to calculate similarity. Therefore, to analyze the incorrect options presented by MLLMs, we

| MLLM | One-Shot | Acc.(%)↑ | ECE |
|---|---|---|---|
| | Cell | **43.24** | 0.2313 |
| Qwen2.5-VL-7B | Science | 42.98 | **0.2190** |
| | Nature | 40.62 | 0.2899 |
| | ACS | 39.27 | 0.2960 |

Table 5: **Model Performance on the Image2Text Task Using One-Shot Examples from Different Publishers.** We evaluated Qwen2.5-VL-7B on the MAC-2025 benchmark using one-shot examples from various publishers, and it demonstrated similar performance..

examined which embedding models these incorrect options originated from and further analyzed whether these models exhibited preferences for specific embedding models. By comparing the distribution of incorrect options, we aim to reveal the differences in how various embedding models select distractors, especially whether certain embedding models tend to frequently choose specific types of distractors when handling the same question.

Figure 7 presents the error rates of several advanced multimodal large language models (MLLMs), including GPT-4o, Gemini-1.5-Pro, Qwen-Max, and the Step series, across four task dimensions. Each model is evaluated against distractors generated using different embedding models (see section 3.2). The results reveal clear performance disparities among models across tasks, particularly along the central average axis, indicating that MLLMs exhibit systematic differences in their susceptibility to confusion under various cross-modal matching tasks. Notably, Qwen2.5-VL-7B consistently shows higher error rates across all tasks, with the highest rates observed under distractors generated by ViT-G and ViT-H, suggesting weaker robustness to semantically similar distractors. In contrast, GPT-4o and Gemini-1.5-Pro demonstrate more stable semantic discrimination.

Overall, error rates are higher in the option domain than in the info domain, suggesting that models face greater difficulty when distinguishing between closely related distractors. Additionally, Text2Image tasks yield slightly higher error rates than Image2Text tasks, indicating that current models still struggle with extracting scientific semantic concepts from text and accurately grounding them in visual content. Interestingly, we also observe greater variation in embedding preferences among different MLLMs on the Text2Image tasks. For instance, in the Text2Image info track, ernie-4.5-8k tends to be more confused by ViT-G-generated distractors, while Qwen2.5-VL-7B is more affected by ViT-H. Gemini-1.5-Pro shows greater susceptibility to ViT-H, whereas Step-1o-Turbo-Vision is more challenged by ViT-G. These findings highlight the potential of using more difficult Text2Image settings, along with diverse distractor generation strategies, to effectively uncover weaknesses in the cross-modal semantic alignment capabilities of different MLLMs—thereby offering a lens into the "hidden language" each model uses to understand visual semantics.

## G   Text-Free MAC-2025 Analysis

Figure 8 presents a representative failure case from our text-free MAC-2025 evaluation, where both GPT-4o and the concurrently released Gemini 2.5-Pro (Google, 2025) misidentified the correct answer. While these state-of-the-art MLLMs successfully recognized the key visual elements in option C—namely, "pills" and a "prescription pad"—demonstrating strong perceptual capabilities, they subsequently reasoned that the image lacked content related to "drug resistance" or "mechanisms of cancer therapy." Instead, both models erroneously inferred that other options depicted cancer cells, leading to incorrect choices.

This highlights a fundamental limitation: despite accurately identifying surface-level visual elements, current MLLMs still struggle to establish deeper semantic associations between those elements and the underlying scientific concepts. Specifically, they fail to recognize that in a biomedical context, the combination of "pills + prescription" may imply "cancer treat-

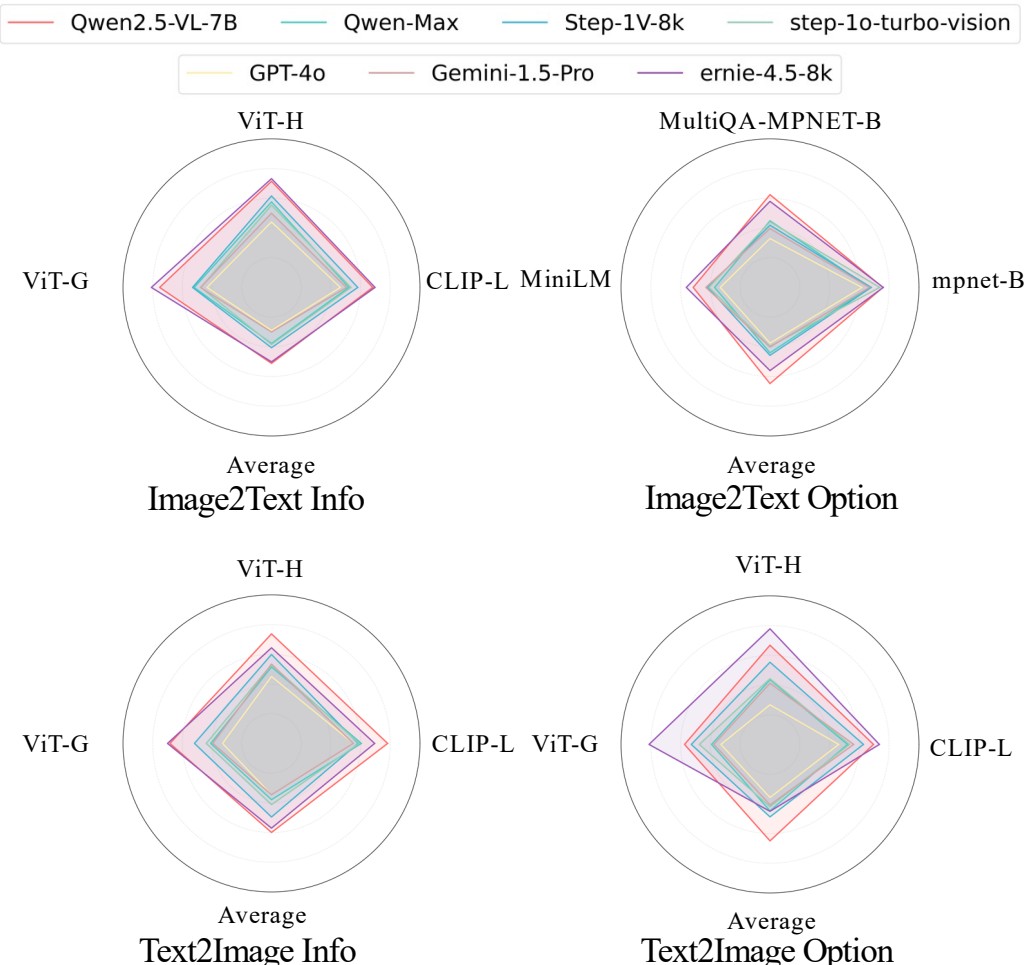

Figure 7: **We performed a comprehensive analysis of error patterns and their origins across all tested MLLMs.** Models exhibit clear differences in their preferences across distractors selected by different embedding models, revealing variability in how each model responds to semantic interference from different embedding spaces.

ment" or "drug resistance." Their understanding remains at the level of visual recognition without fully grasping the latent scientific meaning behind these cues.

This result underscores a broader insight: perceptual ability alone is insufficient for robust multimodal scientific understanding. Bridging the gap between visual recognition and semantic inference requires higher-level reasoning mechanisms that go beyond deriving what it signifies and situating it within broader scientific narratives—toward understanding what it signifies. Not only to recognize, but also to understand—ultimately, to reason.

## H    Models Performance in Description and Deduction

Table 7 presents the performance impact of incorporating different reasoning models into our proposed reasoning-augmented framework, DAD, using GPT-4o as the base model. We evaluate accuracy (Acc.), expected calibration error (ECE), negative log-likelihood (NLL), and root mean square error (RMS) to assess robustness and generalizability. Across both tasks, introducing any reasoning model leads to improved accuracy for GPT-4o. Notably, the combination with Deepseek-R1 yields the highest performance, with accuracy in the info domain increasing from 0.751 to 0.767 and in the option domain from 0.735 to 0.752. These

| MLLM | Image with Text | Prompt with Text | Acc.(%)↑ | ECE↑ |
|---|---|---|---|---|
| GPT-4o | Y | N | 75.1 | 0.0534 |
| Qwen-VL-Max | Y | N | 69.8 | 0.1706 |
| GPT-4o | N | N | 64.8 | 0.0419 |
| Qwen-VL-Max | N | N | 61.6 | 0.1334 |
| GPT-4o | N | Y | 75.5 | 0.0261 |
| Qwen-VL-Max | N | Y | 70.8 | 0.2018 |

Table 6: **Performance Comparison of MLLMs under Different Input Modality Configurations.** We evaluated the performance of two representative models on text-free images of the image2text task under two conditions: with and without supplementing the prompt with text information obtained via OCR. This performance was then compared against the models' baseline results on the unprocessed images. Y denotes 'Yes' and N denotes 'No'.

| MLLM | Reasoning | Image2Text Level | | | | | | | |
|---|---|---|---|---|---|---|---|---|---|
| | | Info Domain | | | | Option Domain | | | |
| | | Acc.(%) | ECE | NLL | RMS | Acc.(%) | ECE | NLL | RMS |
| GPT-4o | / | 75.1 | 0.053 | 1.291 | **0.089** | 73.5 | 0.055 | 1.336 | **0.086** |
| | QwQ-32B | 76.3 | 0.059 | 1.766 | 0.092 | 73.6 | 0.076 | 1.429 | 0.099 |
| | GPT-o3-mini | 75.3 | 0.064 | 1.471 | 0.112 | 74.9 | **0.063** | 1.507 | 0.108 |
| | Deepseek-R1 | **76.7** | **0.051** | **1.243** | 0.092 | **75.2** | 0.064 | **0.938** | 0.101 |

Table 7: **Ablation studies of the reasoning model in DAD.** We evaluated the robustness of our DAD method by substituting its reasoning component with models from various providers and of different parameter scales, consistently demonstrating performance improvements over baseline models.

results highlight the effectiveness of structured reasoning in enhancing GPT-4o's scientific semantic understanding and cross-modal reasoning capabilities.

However, the degree of improvement varies across reasoning models, and some may introduce confidence instability. While QwQ-32B and GPT-o3-mini-high also boost accuracy, their performance in ECE, NLL, and RMS is slightly inferior to that of Deepseek-R1. For instance, QwQ-32B shows a higher ECE of 0.076 in the option domain compared to Deepseek-R1's 0.064, suggesting less stable confidence calibration. Although GPT-o3-mini-high improves accuracy, it exhibits noticeably higher RMS and NLL, indicating greater prediction variance and weaker confidence. These findings suggest that DAD, when paired with more advanced reasoning models such as Deepseek-R1, can further unlock the scientific reasoning potential of MLLMs and lead to more robust multimodal understanding.

# I  Performance of *Live* Data Curation

To further investigate the effectiveness of our *live* data curation strategy, we reconstruct the distractors for all 2,287 samples in MAC-2025 using three advanced embedding models released in 2025, resulting in the 2025-embedding version of the benchmark. We select three representative MLLMs—Qwen2.5-VL-7B, Qwen-Max, and Gemini-1.5-Pro—and compare their performance on benchmarks constructed with both 2023 and 2025 embedding models.

| MLLMs | Dataset Construction | Image2Text Level | | | |
|---|---|---|---|---|---|
| | | Acc(%)↑. | ECE↓ | NLL↓ | RMS↓ |
| Qwen2.5-VL-7B | 2023 Embeddings | 59.7 | **0.055** | **1.856** | **0.095** |
| | 2025 Embeddings | **58.4** | 0.061 | 1.923 | 0.105 |
| Qwen-VL-Max | 2023 Embeddings | 69.8 | 0.171 | 3.054 | 0.210 |
| | 2025 Embeddings | 68.7 | 0.205 | 3.472 | 0.240 |
| Gemini-1.5-Pro | 2023 Embeddings | **72.7** | 0.108 | 5.058 | 0.153 |
| | 2025 Embeddings | 71.8 | 0.128 | 5.567 | 0.165 |

Table 8: **MLLMs' Performance on Benchmarks Generated Using State-of-the-Art Embedding Models.** We reconstructed new distractor items for samples identical to those in MAC-2025 from the MAC dataset using three state-of-the-art embedding models, and evaluated baseline models' performance on the image2text task.

The results in Table 8 show a consistent drop in accuracy across all models when evaluated on the 2025 embedding version: Qwen2.5-VL-7B drops by 1.3%, Qwen-Max by 1.1%, and Gemini-1.5-Pro by 0.9%. All models exhibit a noticeable increase in NLL and RMS, indicating greater prediction uncertainty and reduced confidence stability. ECE values also increase across the board—for example, Qwen-Max's ECE rises from 0.171 to 0.205—suggesting that the newer embedding models generate more semantically similar and harder-to-distinguish distractors, thereby raising the task's difficulty to "former" MLLM.

These findings further validate the necessity and effectiveness of our *live* benchmark design. As embedding models and representation learning techniques continue to evolve, periodically incorporating stronger embedding models to generate distractors helps maintain the benchmark's cutting-edge nature, enabling more accurate assessment of the limits of contemporary MLLMs in scientific semantic modeling and cross-modal understanding.

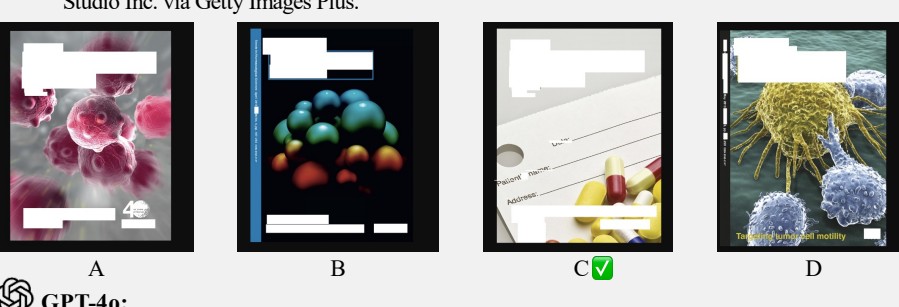

### Text2Image Scientific Understanding Task

I want to ask you the following question: **Which of the following options best describe the cover story?** And you have **the cover story**: Despite the successes in …resistance to these therapies is common and affects disease prognosis and quality of life of patients. Towards eradication of therapy resistance, clinicians and research scientists in different parts of the world have leveraged new, advanced technological tools to gain greater insights of the molecular underpinnings of cancer progression and resistance to treatment. These insights …in tackling therapy resistance in cancer. The June special issue of Trends in Pharmacological Sciences features expert perspectives on the recent developments in the cancer drug resistance field and exciting future avenues of research. Image credit: Lucidio Studio Inc. via Getty Images Plus.

A    B    C ✅    D

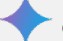 **GPT-4o:**

The image A…. The image C shows several pills scattered on a prescription pad. The pills are of different colors and sizes, arranged randomly near an empty form with spaces for patient information. This suggests a theme related to medication, healthcare, or pharmaceutical practices, highlighting aspects of prescription management or pharmaceutical development. The image D shows a yellow cellular structure resembling a cancer cell with long protrusions, closely surrounded by smaller blueish cells. The image is highly detailed and looks as if it's taken from a microscopic view. The surface of the cells is textured and complex, suggesting a detailed study of cellular interaction or targeting mechanisms. It highlights interactions at a cellular level, possibly addressing issues like cell motility or immune system targeting. **Step-by-step Analysis for Selection**: 1. The provided story focuses on novel cancer therapies, treatment resistance, and molecular insights into cancer progression. 2. Image A shows …Image C represents pharmaceuticals which align with drug therapy but doesn't directly imply cancer-specific advances or resistance themes. 5. Image D specifically features cellular interaction with a possible tumor cell, encompassing a biological basis for how cancer interacts with its environment, aligning with targeting or resistance narratives…

◆ **Gemini 2.5 pro:**

The user wants me to act as a scientific image reader and predict the probability of choosing each option image (A, B, C, D) based on the provided cover story. **Cover Story Analysis:** The cover story discusses cancer therapies (small molecule drugs, biologics). It highlights the problem of therapy resistance in cancer. …**Image Analysis: Image A:** Shows multiple spherical, somewhat irregular pink cells, resembling cancer cells, possibly multiplying or resisting treatment. The background is blurry and suggests a cellular environment… **Image C:** Shows prescription pills and capsules on a prescription form. This relates to therapy (drugs) but doesn't depict the biological aspect (cells, resistance) mentioned prominently in the story. …**Matching Story and Images:** The story is about cancer, specifically therapy resistance and molecular understanding. **Image A:** Directly depicts cells that look like cancer cells, fitting the theme of cancer and potentially representing the cells that become resistant… **Image C:** Focuses only on the therapy (drugs) aspect, missing the biological context of cells and resistance mechanisms…A: High (e.g., 0.7) D: Moderate (e.g., 0.2) B: Low (e.g., 0.05) C: Low (e.g., 0.05) Check sum: $0.7 + 0.2 + 0.05 + 0.05 = 1.0$

Figure 8: **Case study of GPT-4o and Gemini 2.5-Pro on the text-free version of MAC-2025.** Despite being two of the most advanced MLLMs, both models fail to establish the semantic link between the visual elements and the underlying scientific concept. The key scientific concepts from the story are highlighted in the figure, along with the models' judgments on both the ground-truth and distractor options.

