# OpenReview forum: "MAC: A Live Benchmark for Multimodal Large Language Models in Scientific Understanding"
_colmweb.org/COLM/2025/Conference — COLM 2025_

### Official Review · Reviewer_DRu9 · 2025-05-10

**Rating:** 6
**Confidence:** 4
**Ethics Flag:** 1

**Summary:**

This paper introduces MAC, a benchmark for evaluating scientific understanding in MLLMs. It includes a large version with ~25,000 image-text pairs sourced from journal issues in Nature, Science, Cell, and ACS, as well as a smaller version using only issues from 2024 onward to reduce the likelihood of data contamination and better evaluate model generalization. The benchmark is framed as a multiple-choice matching task (text2image and image2text), where the goal is to match covers with their corresponding cover stories and vice versa. To increase difficulty, all options are from the same scientific domain, distractors are selected using embedding similarity, and text elements are removed from cover images to avoid shortcuts. Additionally, the authors propose a Description and Deduction (DAD) strategy, inspired by Vision-R1, which converts images into text and generates a pseudo chain-of-thought before answering. This improves performance across all tested models. They also demonstrate a significant drop in accuracy on the 2024+ subset, highlighting the value of benchmarks with recent data.

**Questions To Authors:**

1) The term “live” is underspecified. How frequently is the benchmark updated? Is this manual or automated? How can results be compared consistently over time if the dataset evolves?

2) Is DAD symmetric across text2image and image2text tasks? It seems that DAD uses 1-shot generation in image2text and 4-shot in text2image. This could explain the higher relative gains in text2image (Figure 4). A short discussion would be helpful.

3) In Appendix Table 5, only one task is shown but twice the same metric. Was text2image omitted by accident?

4) It is interesting that the task is worse with one-shot compared to the baseline. This makes me wonder about the examples used and the fact that few-shot is not being done here.

**Reasons To Accept:**

1) The use of 2024+ journal issues which are comprised of cover-story pairs makes this a forward-looking and precise benchmark for evaluating MLLM performance in scientific understanding. The benchmark is also well-balanced across scientific domains.

2) The modality bridging from Vision-R1 (DAD method) is a clever and practical enhancement for MLLM VQA that outperforms previous approaches.

3) Extensive evaluations are presented, covering various model types, ablations (e.g., bidirectional task, text-free covers), and data curation strategies.

4) MAC is a valuable contribution and could become a standard benchmark for evaluating MLLMs in real-world scientific tasks due to its high-quality covers and stories requiring fine-grained task-specific understanding.

**Reasons To Reject:**

1) No human performance baseline is provided. It would be useful to know whether humans perform reliably on these tasks, especially in combination with the incorporation of the distraction process. Does this also correlate with human confusion?

2) Regarding the removal of text from images, a small experiment using OCR'd text as input (i.e., a "text-only" baseline) could help evaluate whether performance drops are due to missing textual cues or image unnaturalness. Also, details on how the OCR confidence threshold is chosen would be valuable.

3) The paper might benefit from comparing the proposed domain-based candidate selection with a random or cross-domain baseline to highlight the impact of the semantic difficulty design.

4) There are several minor issues. Wrong capitalization of references (e.g., "table 4" → "Table 4"), pluralization (e.g., "Experiment" → "Experiments"), abbreviation inconsistencies, and typos (e.g., Line 246: "mproves" → "improves"). Highlighting best results in tables (e.g., bold or underline) would improve readability.

---

> ### Author Response · Authors · 2025-06-02
>
> Thank you for your valuable feedback and for recognizing the strengths of our work. We truly appreciate your insights. Below, we provide detailed responses to the concerns you raised.
> > R1: No human performance baseline is provided. It would be useful to know whether humans perform reliably on these tasks, especially in combination with the incorporation of the distraction process. Does this also correlate with human confusion?
>
> |       | MAC-2025 random Acc. | MAC-2025 Acc. |
> |:-----:|:--------------------:|:-------------:|
> | human | 0.8                  | 0.7167   |
>
> Thank you for your valuable suggestion. In response to your concern, we selected 20 covers from journals published by different publishers in the MAC-2025 dataset and constructed two types of human evaluation settings: one using random distractor options, and the other using the same embedding-based distractor options as in MAC-2025. We then invited three evaluators, each holding at least a bachelor’s degree, to independently complete the evaluation. The results are presented in the table. As shown, human performance is notably better with random distractor options, while accuracy drops significantly when using the embedding-based distractors. This phenomenon indicates that our distractor design effectively increases the task difficulty, demonstrating the reasonableness and effectiveness of the distractor strategy. In future work, we also plan to release the detailed human evaluation questions and data for further reference and study by the community.
> > R2: Regarding the removal of text from images, a small experiment using OCR'd text as input (i.e., a "text-only" baseline) could help evaluate whether performance drops are due to missing textual cues or image unnaturalness. Also, details on how the OCR confidence threshold is chosen would be valuable.
>
> | MLLM             | Text-Free_Image | Prompt with text in covers | Acc. | ECE|
> |:----------------:|:---------------:|:----------------:|:--------:|:-------:|
> | gpt-4o           | N| N| 0.7508   | 0.0534  |
> | qwen-vl-max | N| N| 0.6977   | 0.1706  |
> | gpt-4o           | Y| N| 0.6480   | 0.0419  |
> | qwen-vl-max | Y| N| 0.6164   | 0.1334  |
> | gpt-4o           | Y| Y| 0.7547   | 0.0261  |
> | qwen-vl-max | Y| Y| 0.7078   | 0.2018  |
>
> Thank you for your valuable suggestion. To address your concern, we have conducted additional experiments to further analyze whether the performance drop in the text-free setting is caused by the unnaturalness of the images. Specifically, we performed a comparative experiment where we used text-free images and supplemented the missing text information in the prompt. As shown in the results table, the performance with text-free images plus text input is almost consistent with the original baseline. This indicates that the performance degradation observed in the text-free setting is primarily due to the absence of textual information rather than the unnaturalness of the images themselves. Regarding the threshold selection for text masking, we randomly sampled 10 cover images from four different publishers and experimented with various thresholds (0.1, 0.2, 0.25, 0.4, 0.5, 0.6, 0.75, 0.8, and 0.9). The final threshold was selected by comprehensively considering both the effectiveness of text removal and the preservation of key visual-semantic elements in the images.
> > R3: The paper might benefit from comparing the proposed domain-based candidate selection with a random or cross-domain baseline to highlight the impact of the semantic difficulty design.
>
> | MLLM                   | Options        | Acc.    |
> |------------------------|----------------|---------|
> | qwen-vl-max| random, nature | 0.8979  |
> | qwen2.5-vl-7b-instruct | random, nature | 0.7943  |
> | qwen-vl-max| clip, nature   | 0.8695  |
> | qwen2.5-vl-7b-instruct | clip, nature   | 0.7574  |
>
> Thank you for the suggestion. In the exploratory phase shown in the table, we compared random and embedding-based distractor selection on a subset of the MAC dataset, and observed significant performance drops with embedding-based distractors, confirming increased difficulty. As for cross-domain selection, since our ground-truth relies on rigorous cover-story pairings within the same journal, expanding to cross-domain could introduce mismatches and compromise ground-truth validity. Therefore, we did not include cross-domain experiments.
>
> The remaining questions will be addressed in the next comment.

---

> > ### Comment · Reviewer_DRu9 · 2025-06-05
> >
> > Thanks, the authors made a very comprehensive review. I am tempted to increase my score (still need a bit of deliberation).

---

> > > ### Comment · Reviewer_DRu9 · 2025-06-06
> > >
> > > As the authors have written a very thorough review, addressing several raised points, I am increasing to 6. However, I also note that there seems to be a long list of points to be addressed in the revision.

---

> ### Author Response · Authors · 2025-06-02
>
> > R4: There are several minor issues. Wrong capitalization of references (e.g., "table 4" → "Table 4"), pluralization (e.g., "Experiment" → "Experiments"), abbreviation inconsistencies, and typos (e.g., Line 246: "mproves" → "improves"). Highlighting best results in tables (e.g., bold or underline) would improve readability.
>
> Thank you for pointing this out. We sincerely apologize for these oversights. We have carefully proofread the manuscript, corrected the formatting and typographical issues, and standardized the expressions. Following your suggestion, we have also bolded the best results in all tables to improve readability. Thank you again for your thorough review and valuable suggestions.
> > Q1: The term “live” is underspecified. How frequently is the benchmark updated? Is this manual or automated? How can results be compared consistently over time if the dataset evolves?
>
> Thank you for your question! We plan to automatically collect the latest content from journals each year to construct and release updated versions of our benchmark. Each year’s dataset will be based on new content published by the same journals, ensuring continuity and consistency in data sources. This design allows our benchmark to continuously evolve over time, maintaining data stability while reflecting the latest research trends and changes in model capabilities.
> > Q2: Is DAD symmetric across text2image and image2text tasks? It seems that DAD uses 1-shot generation in image2text and 4-shot in text2image. This could explain the higher relative gains in text2image (Figure 4). A short discussion would be helpful.
>
> Thank you for your question. Our DAD framework consists of two modules. The first module is a Multimodal Large Language Model (MLLM), which takes an image as input and outputs a description of the image. For both the image2text and text2image tasks, DAD adopts a consistent processing procedure: all images are input into the MLLM at once to generate corresponding descriptions. It is important to emphasize that this process is aligned with the standard inference approach of using MLLMs directly, ensuring fairness and consistency between the two tasks at the MLLM stage. In the second module, the reasoning model, one description is also used as input for both the image2text and text2image tasks, thereby ensuring fair comparison across tasks within the DAD framework.
> > Q3: In Appendix Table 5, only one task is shown but twice the same metric. Was text2image omitted by accident?
>
> Thank you for your careful feedback. We sincerely apologize for the lack of detailed annotation in Table 5. The metrics on the left side of the table correspond to evaluation results under the Info Domain setting, while those on the right side correspond to results under the Option Domain setting, both evaluated on the image2text task. Considering the balance between experimental resources and computational costs, we chose to conduct this analysis only on the image2text task for this round of experiments, with the primary goal of verifying the performance trends of different reasoning models within the DAD framework. In future work, we also plan to extend and validate these experiments on the text2image task to further enhance the completeness and persuasiveness of our study.
> > Q4: It is interesting that the task is worse with one-shot compared to the baseline. This makes me wonder about the examples used and the fact that few-shot is not being done here.
>
> | MLLM| One-Shot| Acc.| ECE|
> |:-------------:|:--------:|:-----------:|:-----------:|
> | Qwen2.5-VL-7B|Cell| 0.432| 0.231|
> |Qwen2.5-VL-7B|Science|0.430| 0.219|
> |Qwen2.5-VL-7B|Nature|0.406| 0.290|
> |Qwen2.5-VL-7B|ACS|0.393| 0.296|
>
> Thank you for your question. In our one-shot experiments, we selected four examples from four different publishers. As shown in the table, the performance of each one-shot example is generally comparable. Therefore, in the paper, we chose to present the result from the Cell publisher, which achieved the best performance. Regarding the few-shot setting, considering that the one-shot results showed relatively limited improvements and that few-shot experiments would significantly increase the training time and computational cost, we did not conduct few-shot experiments in this work. In future studies, we plan to further explore performance variations under few-shot settings to more comprehensively evaluate the model’s behavior under different conditions.
>
> Thank you again for your valuable feedback and thoughtful suggestions. Your insights have been instrumental in helping us refine and strengthen our work. We believe the revisions and additional analyses we have provided address your concerns, and we look forward to continuing to improve the benchmark and contribute to the broader research community.

---

### Official Review · Reviewer_3gnu · 2025-05-10

**Rating:** 7
**Confidence:** 5
**Ethics Flag:** 1

**Summary:**

The paper presents a valuable contribution to evaluating scientific reasoning in MLLMs with a well-designed benchmark and an effective method to improve performance. The summary of strong points are:

1. The paper introduces MAC-2025, a benchmark using scientific journal cover images (25,000+ image-text pairs) for evaluating multimodal models' scientific reasoning capabilities.
2. Proposes a "Description and Deduction" inference time framework to improve model performance by connecting visual cues with language-based reasoning.
3. Creates a continuously adaptive benchmark framework that can evolve with advancing MLLM capabilities.
4. Rigorous evaluation by including text-free image analysis to ensure models actually understand visual content rather than relying on textual cues.

The summary of weak points are:
1. While scientific cover images are valuable, they represent only one aspect of scientific understanding. so title is slightly misrepresenting. Need more strong motivation of why this is an important problem to tackle and cant be handled by other scientific multimodal benchmarks
2. The paper doesn't fully address how the benchmark measures different levels of scientific reasoning complexity. for e.g., in benchmarks like ALgoPuzzle VQA, VisualPuzzles etc.

**Questions To Authors:**

1. Did authors thought about providing reasoning traces along with groundtruth response given nowdays there is so much interest in reinforcement post training methods
2. Could you elaborate on how the "Description and Deduction" framework might be applied to other scientific reasoning tasks like MathVista  ?
3. What metrics do you use to differentiate between perception-understanding and cognition-reasoning abilities of MLLM?
4. How did you validate the accuracy/correctness of groundtruth pairs ?
5. Can you explain the variation of results due to different reasoning model/metrics

**Reasons To Accept:**

1. Focuses on a novel area (scientific journal cover) of scientific multimodal use cases,
2. Provides a comprehensive and an adaptable framework for continuous assessment as models evolve
3. Also proposes a new method" Description and Deduction"to improve the performance on the benchmark, which shows promising performance improvements.
4. Evaluation methodology is thoughtful, including text-free testing

**Reasons To Reject:**

1. Not sure which aspect of Multimodal Language models, this benchmark is going to test. This is just like any other image-text caption task. What is so unique about this benchmark dataset with respective to MLLM evaluation.
2. Insufficient details about the dataset's distribution across scientific disciplines and potential biases
3. Limited discussion of how this benchmark compares with other scientific reasoning tasks.
4. How did the authors validate the accuracy/correctness of groundtruth pairs. Explicit reliance on inherent correctness of the journal cover-image pairs as published by scientific journals but would be better to do human validation on certain subset  to showcase the correctness. may need to report some kind of inter annotator score to handle subjectivity.
5.  Doesn't fully explore/explain why certain reasoning models perform better than others within the DAD framework and there is no consistent story with some of the results.

---

> ### Author Response · Authors · 2025-06-02
>
> Thank you for your valuable feedback and for recognizing the strengths of our work. We truly appreciate your insights. Below, we provide detailed responses to the concerns you raised.
> > R1: Not sure which aspect of Multimodal Language models, this benchmark is going to test. This is just like any other image-text caption task. What is so unique about this benchmark dataset with respective to MLLM evaluation.
>
> Thank you for your question. First, our benchmark primarily focuses on evaluating the scientific understanding capabilities of multimodal large language models (MLLMs), presenting a set of challenging bidirectional comprehension tasks targeting the current limits of model performance. Second, we innovatively leverage the evolving nature of scientific content to design a benchmark with live data and live data curation mechanisms, ensuring that the evaluation content consistently aligns with the cutting edge of human knowledge. Compared to traditional static benchmarks, which tend to become less effective as model capabilities improve, our approach effectively mitigates the problem of rapid data memorization and maintains the benchmark’s difficulty and evaluation capability over time.
> > R2: Insufficient details about the dataset's distribution across scientific disciplines and potential biases.
>
> Thank you for your reminder. During the data collection process, we have covered multiple major disciplines, including natural sciences, life sciences, materials science, and medicine. Our data sources encompass top academic journals such as Nature, Science, Cell, and ACS, as shown in Figure 2. To ensure reasonable and comprehensive domain coverage, we leveraged the broad disciplinary settings of these journals and referred to their official classification systems (e.g., Chemistry, Biology, Medicine), maintaining good alignment between our dataset and mainstream research fields. Meanwhile, in response to your concern about potential biases, we will supplement the final version with more detailed data statistics, including a complete list of the 188 journals in our dataset and their respective sample counts. This will help readers gain a comprehensive understanding of the dataset’s distribution and potential domain biases, thereby further enhancing the transparency and credibility of our benchmark.
> > R3: Limited discussion of how this benchmark compares with other scientific reasoning tasks.
>
> Thank you for your question. We highly acknowledge the important contributions of other scientific reasoning benchmarks, such as MathVista and ScienceQA, in advancing the field. It is important to clarify that our work focuses specifically on scientific vision-language understanding and aims to build a benchmark with a live updating mechanism. Therefore, its focus does not directly overlap with existing reasoning benchmarks, and it is not intended to compete with these tasks. Instead, we view our benchmark as complementary to existing efforts. In future work, we plan to further extend our experimental setup by incorporating comparisons with mainstream scientific reasoning benchmarks to provide a more comprehensive and systematic evaluation of models’ scientific reasoning and understanding capabilities. Thank you again for your valuable suggestions, which are highly helpful for improving our future work.
> > R4: How did the authors validate the accuracy/correctness of groundtruth pairs. Explicit reliance on inherent correctness of the journal cover-image pairs as published by scientific journals but would be better to do human validation on certain subset to showcase the correctness. may need to report some kind of inter annotator score to handle subjectivity.
> Q4: How did you validate the accuracy/correctness of groundtruth pairs?
>
> Thank you for your question. In this study, we indeed primarily rely on the cover-story pairings published by leading journals as the ground truth. It is important to note that the selection process for journal covers and their corresponding stories typically undergoes a rigorous review procedure. For example, in Cell (https://www.cell.com/cell/authors), the cover image and its caption must pass multiple stages of scrutiny, including editorial pre-assessment, author submission, internal editorial review and selection, and peer review, ensuring a high degree of scientific and artistic alignment between the cover and the associated story. Therefore, within the same journal, the cover and story naturally possess a high-quality and trustworthy pairing. Moreover, to further avoid the risk of better matching options across different journals, we restrict the distractor candidates to come only from the same journal in our task design, thereby maximizing the validity and fairness of the evaluation. In future work, we also plan to incorporate human validation or additional human evaluation to further enhance the reliability of our benchmark data.
>
> The remaining questions will be addressed in the next comment.

---

> > ### Comment · Reviewer_3gnu · 2025-06-07
> >
> > Thanks to the authors for the detailed response and addressing some of the concerns related to the data distribution, accuracy of groundtruth pairs and comparison with other scietific benchmarks. I am not still not convinced with the novelty of the benchmark in terms of its utility in testing which abilities of an vision language models, beside being live. Whether its a scientific image understanding (encoders), reasoning, alignment, etc. It would be nice to clarify on this.

---

> > > ### Author Response · Authors · 2025-06-11
> > >
> > > Thank you very much for your valuable feedback and further questions. Your question, 'Besides being 'live,' what specific capabilities of a model does this benchmark test?' addresses the core of the issue. The uniqueness of our MAC benchmark lies in its use of specialized scientific semantic materials to challenge MLLMs' capabilities in deep scientific visual understanding and cross-modal concept alignment.
> > >
> > > Unlike everyday images, scientific journal covers often employ highly abstract and artistic visual language to convey complex scientific discoveries. For instance, an image might superficially depict pills and a prescription pad, but its deeper scientific concept could be "cancer treatment" or "drug resistance." The MAC benchmark does not just test whether a model can "see" the basic elements in an image, but challenges its ability to "understand" the abstract and extended meanings of these elements within a specific scientific context.
> > >
> > > Simultaneously, the core task of the MAC benchmark is bidirectional image-text matching. This task requires the model to match a cover image, full of abstract and artistic visual expressions, with a "cover story" that explains the cutting-edge scientific discovery behind it in complex and information-dense language. This far exceeds the difficulty of aligning a picture of a "cat" with the word "cat" in a typical image-text dataset. It demands that the model accurately capture and align the shared, highly abstract core scientific concepts across both modalities, posing a rigorous test of the model's cross-modal alignment capabilities for scientific concepts.
> > >
> > > Your suggestions are very insightful. Building on our current work, we plan to design more fine-grained evaluation dimensions and metrics. Such an approach will allow us to deconstruct a model's overall performance in the matching task into specific scores on sub-tasks like visual understanding and concept alignment. Consequently, it will enable a clearer quantification of the model's specific deficiencies in achieving true scientific understanding.
> > >
> > > Thank you again for your valuable feedback. We hope this response more clearly elucidates the uniqueness and value of our work.

---

> ### Author Response · Authors · 2025-06-02
>
> > R5: Doesn't fully explore/explain why certain reasoning models perform better than others within the DAD framework and there is no consistent story with some of the results.
>
> Thank you for your question. We understand that your concern lies in why the performance gains brought by the DAD framework vary across different models. The primary reason for this difference stems from the inherent capability gap between the base MLLMs. Specifically, weaker MLLMs are generally able to capture basic visual-semantic information from the cover images but struggle to establish deep connections between visual elements and scientific concepts. As illustrated in the case study in Figure 8, such models often fail to accurately complete scientific understanding tasks without additional reasoning support. By introducing the reasoning model in the DAD framework, these models can significantly enhance their ability to bridge visual semantics with scientific concepts, leading to notable performance improvements. In contrast, stronger MLLMs already possess a certain level of visual-semantic understanding and association capabilities. For these models, the reasoning model primarily performs inference based on the already high-level abstract descriptions generated by the MLLM, resulting in a relatively smaller improvement margin. Therefore, the variance in gains reflects the different starting points of the underlying models.
> > Q1: Did authors thought about providing reasoning traces along with groundtruth response given nowdays there is so much interest in reinforcement post training methods
>
> Thank you for your question. We fully acknowledge the important role of reasoning traces in enhancing model interpretability and supporting reinforcement learning (RL) post-training. During our experiments, we have preserved the complete reasoning processes generated by the reasoning models based on the descriptions produced by different MLLMs. In our future dataset release, we plan to include these reasoning traces alongside the benchmark, providing a valuable resource for the community to facilitate further research on reinforcement learning and the enhancement of reasoning capabilities.
> > Q2: Could you elaborate on how the "Description and Deduction" framework might be applied to other scientific reasoning tasks like MathVista ?
>
> Thank you for your attention and question! The DAD framework is fundamentally a general-purpose reasoning enhancement strategy and is not tied to any specific task domain by design. Its two-stage mechanism — first generating structured descriptions and then performing reasoning based on these descriptions — makes it particularly well-suited for tasks that require detailed understanding of visual and textual elements along with complex reasoning. For scientific reasoning tasks like MathVista, the DAD framework can similarly leverage the descriptive capabilities of MLLMs to enhance image understanding through the reasoning model. In the future, we plan to further evaluate the transferability and generalization of the DAD framework on benchmarks such as MathVista, and related work is already actively in progress.
> > Q3: What metrics do you use to differentiate between perception-understanding and cognition-reasoning abilities of MLLM?
>
> Thank you for your question, and we sincerely apologize for any confusion caused by the unclear expression. The concepts of perception-understanding and cognition-reasoning that you mentioned are indeed discussed in the Related Works section; however, it is important to clarify that they are not the core focus of this paper. Our primary goal is to propose a live benchmark designed to continuously evaluate the scientific understanding capabilities of models. Regarding perception-understanding and cognition-reasoning, we acknowledge that we have not introduced explicit evaluation metrics for these concepts in the current version. Their inclusion was intended to illustrate the two-stage process that MLLMs typically follow when performing scientific understanding tasks: first, leveraging perception capabilities to interpret visual elements, and subsequently, utilizing cognition capabilities to establish connections between visual information and scientific concepts. Furthermore, through the DAD framework, we aim to enhance and assess model capabilities during the reasoning phase, with performance improvements before and after reasoning serving as an indirect reflection of a model’s information integration and reasoning depth when tackling complex tasks. Thank you again for your careful and constructive feedback. We will pay greater attention to the precision of terminology and descriptions in future revisions to avoid any potential misunderstandings.
>
> The remaining questions will be addressed in the next comment.

---

> ### Author Response · Authors · 2025-06-02
>
> > Q5: Can you explain the variation of results due to different reasoning model/metrics
>
> Thank you for your question. The primary reason for the performance variations under different reasoning models lies in their differing abilities to process the descriptive inputs generated by the DAD framework. Larger models, benefiting from greater scale and richer training data, possess stronger capabilities in constructing reasoning chains and integrating information, thus achieving more significant performance improvements within the DAD framework. In contrast, smaller models face bottlenecks when handling complex reasoning demands, resulting in limited gains or even performance degradation on certain tasks. Differences in text comprehension, reasoning depth, and detail-capturing abilities among reasoning models contribute to the observed variations in evaluation metrics such as accuracy and recall. In future work, we plan to conduct more systematic studies to further explore the relationship between model scale and reasoning capabilities.
>
> Once again, we sincerely thank the reviewer for the thoughtful and detailed feedback. Your comments have been extremely valuable in helping us clarify our contributions and in guiding future improvements to our work.

---

### Official Review · Reviewer_LGGS · 2025-05-12

**Rating:** 6
**Confidence:** 5
**Ethics Flag:** 1

**Summary:**

This paper utilizes the image covers from the famous journals like Nature, Science, and Cell to assess the scientific understanding abilitites of MLLMs. They frame this as a classification task and found that current MLLMs still struggle with this task, potentially due to the lack of scientifc understanding data. They also promise to build a live benchmark to reflect the zero-shot ability of sota models to avoid data contamination. besides, they propose a lightweight inference-time approach to enhance its ability.

**Questions To Authors:**

I noticed another highly relevant work MMSCi (https://arxiv.org/abs/2407.04903) is also curating images from journals for text2image, image2text tasks, but not cited.


afaik, many images on NSC journals are licensed. how can you open-source them?

**Reasons To Accept:**

1. The dataset is very meaningful for scientific understanding of MLLMs, especially with the saturation of current benchmarks.

2. The idea of live benchmark is a very good practice to avoid data contamination, thus reflecting the real ability of models.

**Reasons To Reject:**

1. There are limited models tested, though I understand there is budge limit for testing all models. But testing more open-source models shouldn't have any problems.

2. There is a lack of comprehensive evaluation of human performance, both non- and professional human experts.  Without this, it's hard to interpret the uppbound of model's performance.

---

> ### Author Response · Authors · 2025-06-02
>
> Thank you for your valuable feedback and for recognizing the strengths of our work. We truly appreciate your insights. Below, we provide detailed responses to the concerns you raised.
> > R1: There are limited models tested, though I understand there is budge limit for testing all models. But testing more open-source models shouldn't have any problems.
>
> Thank you for your valuable suggestion. Due to constraints on computational resources and associated costs, we currently conducted experiments on ten models, which indeed limits broader coverage — an aspect we acknowledge with some regret. The primary focus of this work is to demonstrate the discriminative power and challenge level of the benchmark. Given that, in the field of multimodal large language models (MLLMs), closed-source models generally outperform open-source models across various benchmarks, we prioritized evaluating widely recognized, state-of-the-art closed-source models to better showcase the complexity and rigor of the MAC-2025 benchmark. Nevertheless, we also included representative open-source models from the Qwen series to reflect the latest progress and potential of the open-source community. In future work, we aim to expand model coverage as resources permit, to further enhance the comprehensiveness and reference value of the benchmark.
> > R2: There is a lack of comprehensive evaluation of human performance, both non- and professional human experts. Without this, it's hard to interpret the uppbound of model's performance.
>
> | Evaluator | MAC-2025 Acc. |   |   |   |   |   |
> |-----------|---------------|---|---|---|---|---|
> | human1    | 0.85          |   |   |   |   |   |
> | human2    | 0.55          |   |   |   |   |   |
> | human3    | 0.75          |
> Thank you for your insightful comment. To address this concern, we constructed a human evaluation task using 20 covers selected from different journals within the MAC-2025 dataset, following the same option-setting methodology used for the MLLM experiments. We invited three evaluators with at least a bachelor’s degree to independently complete the evaluation. The results are presented in the table and show that human performance on the task is relatively strong, further validating the reasonableness of the task design and demonstrating the inherent challenge and necessity of scientific understanding tasks for MLLMs. In future work, we also plan to publicly release the specific human evaluation questions and data to facilitate further community research and reference.
>
> > Q1: I noticed another highly relevant work MMSCi (https://arxiv.org/abs/2407.04903) is also curating images from journals for text2image, image2text tasks, but not cited.
>
> Thank you for your reminder. We sincerely apologize for the oversight in not citing this relevant work. We will add the appropriate citation to MMSCi in the camera-ready version of our paper. Thank you again for your careful review and valuable suggestion.
>
> > Q2: afaik, many images on NSC journals are licensed. how can you open-source them?
>
> Thank you for your question. As shown in Appendix A, all cover stories and cover images in our dataset are sourced exclusively from open-access channels. Throughout the data collection and release process, we strictly adhered to the license agreements provided by the respective publishers. Furthermore, the dataset is intended solely for scientific research purposes and is explicitly prohibited from any commercial use. We appreciate your attention to data compliance issues and your thoughtful reminder.
>
> Once again, we sincerely thank the reviewer for the thoughtful and detailed feedback. Your comments are highly valuable and have greatly helped us refine and strengthen our work.

---

> > ### Comment · Reviewer_LGGS · 2025-06-11
> >
> > thanks for the response and I will keep my score

---

### Decision · Program_Chairs · 2025-07-08

**Decision:**

Accept

**Comment:**

The authors introduce a new, living benchmark for vision+language models in understanding scientific magazine cover images coupled with the corresponding stories. There are two retrieval tasks (cover -> story; story -> cover). Evaluation is accuracy and ECE; and models leave room for improvement. A reasoning-based augmentation improves performance.

Reviewers were generally positive about this task and the work overall.

- LGGS asked for a human estimate of performance, which was provided in response;
- some reviewers questioned the novelty of yet another image/text matching task but, the authors convincingly argue that the content itself requires different types of reasoning compared to, e.g., caption retrieval
- the authors did a great job in response, but some reviewers note that the magnitude of promised revisions is high so there's some chance that the holistic final product is somewhat more different than the submission c.f. a median case

Overall, I recommend acceptance --- a new benchmark thoroughly evaluated with headroom is difficult to argue against.


(One comment I'd make personally: the title mentions "scientific understanding", but, this particular flavor of scientific understanding is quite specific to understanding the covers of magazines).